# Stress Relaxation Behaviour Modeling in Rigid Polyurethane (PU) Elastomeric Materials

**DOI:** 10.3390/ma16083156

**Published:** 2023-04-17

**Authors:** Paweł Zielonka, Krzysztof Junik, Szymon Duda, Tomasz Socha, Krzysztof Kula, Arkadiusz Denisiewicz, Kayode Olaleye, Wojciech Macek, Grzegorz Lesiuk, Wojciech Błażejewski

**Affiliations:** 1Department of Mechanics, Materials Science and Biomedical Engineering, Faculty of Mechanical Engineering, Wroclaw University of Science and Technology, Smoluchowskiego 25, 50-370 Wrocław, Poland; 2Strongflex.eu Company, Zakrzowska 21, 51-318 Wrocław, Poland; 3Faculty of Civil Engineering, Architecture and Environmental Engineering, University of Zielona Góra, ul. Prof. Z. Szafrana 1, 65-516 Zielona Góra, Poland; 4Faculty of Mechanical Engineering and Ship Technology, Gdańsk University of Technology, 11/12 Gabriela Narutowicza Street, 80-233 Gdańsk, Poland

**Keywords:** polyurethane, elastomers, relaxation, experimental analysis

## Abstract

Polyurethane (PU) has been used in a variety of industries during the past few years due to its exceptional qualities, including strong mechanical strength, good abrasion resistance, toughness, low-temperature flexibility, etc. More specifically, PU is easily “tailored” to satisfy particular requirements. There is a lot of potential for its use in broader applications due to this structure–property link. Ordinary polyurethane items cannot satisfy people’s increased demands for comfort, quality, and novelty as living standards rise. The development of functional polyurethane has recently received tremendous commercial and academic attention as a result. In this study, the rheological behavior of a polyurethane elastomer of the PUR (rigid polyurethane) type was examined. The study’s specific goal was to examine stress relaxation for various bands of specified strains. We also suggested the use of a modified Kelvin–Voigt model to describe the stress relaxation process from the perspective of the author. For the purpose of verification, materials with two different Shore hardness ratings—80 and 90 ShA, respectively—were chosen. The outcomes made it possible to positively validate the suggested description in a variety of deformations ranging from 50% to 100%.

## 1. Introduction

Due to its distinctive combined effect of unusual properties, such as outstanding mechanical strength, great abrasion resistance, toughness, low-temperature versatility, resistance to corrosion, ease of processing, etc., polyurethane (PU), which was first produced by a German professor (Professor Dr. Otto Bayer) and his colleagues in the 1940s [1], has been used in a very broad range of commercial and industrial fields. The urethane group (-NHCOO-), which is formed by the reaction of isocyanate (-NCO), polyols (-OH), and other additives, is the fundamental repeating unit in PUs [2]. The two building components that make up segmented polyurethanes are macrodiol (polyether or polyester diol) for the soft segment and diisocyanate and low-molecular-weight chain extenders or crosslinkers for the hard segment [3]. It has good chemical and mechanical qualities and is used extensively in a variety of industries, including the leather, printing, and automotive industries [4,5,6]. Thermoset polyurethane elastomers are one of the most important subgroups of polyurethane elastomers and can be used in the aerospace industry and other sectors [7,8,9] due to their advantages of good topological structure stability, chemical resistance, wear resistance, and thermal stability [6,10,11]. In 1962, researchers began to realize the significance of the study of rheology in elastomers and, more specifically, in rubber. The rheological phenomena that occur in elastomers under continuous deformation were described by Gent [12]. There are two circumstances and occurrences that are typical from an engineering perspective. When there is sufficient thermal energy to enable chain motion, the immiscibility of the hard and soft segments, which is driven by thermodynamics, results in microphase segregation at lower temperatures. As a result, a two-phase morphology is produced, with glassy or semi-crystalline hard domains acting as both reinforcing fillers and physical cross-links between the soft regions of the rubbery matrix. In terms of their overall polymer content and degree of phase separation, these materials vary from hard rubbers to elastomers at room temperature. Their viscous qualities are linked to irreversible deformation (flow), whose intensity continuously rises over time in response to a given force value. A material’s viscosity is a measurement of the flow resistance it poses. Elastic properties are related to reversible deformations, which disappear as soon as the cause (force) ceases. Thus, elasticity measures a material’s ability to immediately return to its original form after stress relief. The primary method of describing viscoelasticity is a linear combination of viscous and elastic properties. Linear stress and strain relationships are the outcome (or their time derivatives). The coefficients of linear relationships (elastic modulus and viscosity) are constant, i.e., independent of strain. Limiting deformations to modest values is a requirement for reproducing the characteristics of genuine viscoelastic bodies by linear stress–strain relationships [13]. The rheology of polymers is an essential issue in evaluating their behavior and stability during use. The importance of the topic of rheology in elastomers—more specifically, in rubber—was recognized as early as the study in [12], which described the rheological phenomena occurring in elastomers under constant deformation. From an engineering point of view, two typical situations and phenomena occur. The first one is a phenomenon—creep—and stress relaxation. A prerequisite for capturing the properties of real viscoelastic bodies by linear stress–strain relationships is limiting deformations to small values. Particularly in the automotive industry, a lot of effort has been put into rubber materials—or rubber-like materials—for rheological studies of passive damping systems. In general, the mathematical modeling of the rheological behavior of polymeric materials can be performed using a combination of elementary Voigt and Maxwell models. Many works address the topic of polymer rheology much more frequently than metallic materials [14,15,16,17,18,19]. Particularly in the automotive industry, a lot of effort has been put into rubber/composite materials for rheological studies of passive damping systems, such as the studies in [18,19,20,21,22,23,24].

The presented brief review of the literature and the analysis of approaches shows that, so far, no good model/approach to stress relaxation analysis for the materials under study has been defined. While for related material groups—if we can consider rubbers or polyurethane rubbers as such—there are some solutions, it is noted that there is a significant lack of analysis of materials specified within the framework of this dissertation. Especially for large deformations > 50%, important limitations of the models were observed. The study’s specific goal was to examine stress relaxation for various bands of the specified strains.

## 2. Materials and Methods

For evaluation, two sets of polyurethane materials with hardness values of 80 ShA and 90 ShA were chosen. Duroplastic polyurethane was created by casting in an automated molding system, which combined various compounds to create a compound with properties that meet the designer’s specifications—as was presented in previous authors’ study [21]. The final step was to pour the mixture into a hot mold, after which it was placed in an oven to cure. The required level of hardness determined the curing time. The component was then removed from the mold afterwards. For polyurethane materials, hardness measurements were carried out using Shore scale durometers. The basic mechanical properties of the tested materials—as described in paper [21]—are listed in Table 1.

Stress relaxation rheological tests were conducted on specially designed PS (pure shear)-type specimens, as shown in Figure 1.

Considering the homogeneous state of deformation in the central part of the specimen, it was important to design a clamping system for the testing machine that would allow uniform axial load transfer. The gripping system is shown in Figure 2. The stress relaxation test consisted of monotonically loading the specimen to a given strain level and then recording the change in force as a function of time until it stabilized. For this purpose, it was necessary to calibrate the relationship between displacement and strain for both types of specimens of different hardness. The calibration tests were carried out on an INSTRON testing machine equipped with a video-extensometer enabling the evaluation of strain changes as a function of displacement. Proper tests were carried out with controlled displacement on an MTS858 Bionix testing machine. The test stand is shown in Figure 3. The results of the calibration curves are shown in Figure 4; as expected for such prepared specimens regardless of the hardness of the material, the relationship between the strain and displacement of the machine crosshead was similar. In the next part of the study, relaxation curves for both types of material were recorded for the selected strain intervals with steps of 25%.

To avoid Mullin’s effect, all specimens before proper tests were pre-cycled (1000 cycles) with a frequency equal to *f* = 1 Hz and displacement range of 5 mm, which corresponded to an approximate initial strain level of 25%. All experiments were conducted using an MTS858 Bionix servo-hydraulic testing machine. An exemplary force response of both the 80 ShA and 90 ShA PU is shown in Figure 5.

## 3. Modeling and Experimental Results

During the tests, the time, force, and grip-to-grip distance signals were measured. However, for some calculation reasons and for the numerical analysis, proper strain measurement was required. For this purpose, additional tests were performed on the Instron tensile machine equipped with a video-extensometer. During this experiment, a simple relationship between the grip-to-grip distance and local strain was established. Representative calibration curves for the 80 ShA and 90 ShA samples are plotted in Figure 6.

Based on such a prepared setup, the properly conditioned specimens were subjected to various initial strains from 25–100% (80 ShA) and from 25–150% (90 ShA). Next, the change in the force as a function of time was recorded to represent the rheological response of the material. The results are shown in Figure 7a,b.

In the one-dimensional elementary Maxwell model, spring and dashpot elements are assembled in series. Under a constant strain, ε, the time-dependence of the stress, σ, response can be expressed as:(1)$$\frac{d\varepsilon }{dt}=\frac{1}{E}\cdot \frac{d\sigma }{dt}+\frac{\sigma }{\eta }$$
where:*E*—elastic constant (such as Young’s modulus);*η*—the ratio of the viscosity;*ε*—applied strain;*σ*—stress;*t*—time.

In the case of the relaxation test, the strain was constant, so this can be written as follows:(2)$$\frac{1}{E}\cdot \frac{d\sigma }{dt}=-\frac{\sigma }{\eta }$$

Solving the above differential equation with the known initial boundary conditions gives the following:(3)$$\sigma_{0}\left(t\right)=\sigma_{0}e^{-\frac{tE}{\eta }}$$

After dividing by the strain, the so-called relaxation modulus is given as follows:(4)$$E\left(t\right)=\frac{\sigma_{0}}{\varepsilon }e^{-\frac{tE}{\eta }}$$

It is worth noting that in the case of ideally elastic materials, a description using the Voigt model of stress relaxation is impossible, such as this one, due to the parallel connection of elastic and damping components:(5)$$\varepsilon \left(t\right)=\varepsilon \left(t_{0}\right)=\frac{\sigma_{0}}{E}\Rightarrow \sigma \left(t\right)=\sigma_{0}=E\varepsilon$$

The literature analysis showed that the physical and rheological behavior of polymers is more complicated and requires combining the Voigt and Maxwell models:(6)$$J\left(t\right)=\frac{1}{E_{0}}+\frac{1}{E_{1}}\cdot \left(1-e^{-\frac{E_{1}\cdot t}{\eta_{1}}}\right)+\frac{1}{E_{2}}\left(1-e^{-\frac{E_{2}\cdot t}{\eta_{2}}}\right)$$
where:*E_0_*, *E*_1_, *E*_2_, *η*_1_, and *η*_2_ are model parameters.

For such a five-parameter representation, all the strains can be divided as follows:(7)$$\varepsilon =\varepsilon_{0}+\varepsilon_{1}+\varepsilon_{2}$$

Based on an initial linear relationship (for small strains levels), it can be concluded that:(8)$$\varepsilon_{0}=\frac{\sigma }{E_{0}}$$

In the Kelvin–Voigt model, the differential form of the constitutive equation takes the following form [19]:(9)$$\sigma =E\cdot \varepsilon +\eta \cdot D_{t}\varepsilon$$
where:*D_t_* represents differentiation operator.

Based on the above:(10)$$\sigma =E_{1}\cdot \varepsilon_{1}+\eta_{1}\cdot D_{t}\varepsilon_{1}$$
(11)$$\sigma =E_{2}\cdot \varepsilon_{2}+\eta_{2}\cdot D_{t}\varepsilon_{2}$$

$\varepsilon_{1}$ and $\varepsilon_{2}$ can be directly calculated from Equations (10) and (11). Next, this can be substituted into (7), and the final form of the five-parameter model can be expressed as:(12)$$p_{0}\cdot \sigma +p_{1}\cdot D_{t}\sigma +p_{2}\cdot D_{t}^{2}\sigma =q_{0}\cdot \varepsilon +q_{1}\cdot D_{t}\varepsilon +q_{1}\cdot D_{t}^{2}\varepsilon$$
whereby:(13)$$p_{0}=E_{0}\cdot E_{2}+E_{1}\cdot E_{2}+E_{0}\cdot E_{1},\;p_{1}=\left(E_{0}+E_{1}\right)\cdot \eta_{2}+\left(E_{2}+E_{0}\right)\cdot \eta_{1},\;p_{2}=\eta_{1}\cdot \eta_{2}$$
(14)$$q_{0}=E_{0}\cdot E_{1}\cdot E_{2},\;q_{1}=E_{0}\cdot \left(E_{1}\cdot \eta_{2}+E_{2}\cdot \eta_{1}\right),\;q_{2}=E_{0}\cdot \eta_{1}\cdot \eta_{2}$$

For the relaxation mode, the deformation can be finally expressed as:(15)$$\varepsilon (t)=\varepsilon_{0}\cdot H(t)$$
where:*ε*_0_—strain for *t*_0_ = 0, where *H*(*t*) represents Heaviside’s function.

Based on (12):(16)$$p_{0}\cdot \sigma +p_{1}\cdot D_{t}\sigma +p_{2}\cdot D_{t}^{2}\sigma =q_{0}\cdot \varepsilon_{0}\cdot H\left(t\right)+q_{1}\cdot \varepsilon_{0}\cdot \delta \left(t\right)+q_{2}\cdot \varepsilon_{0}\cdot D_{t}\delta \left(t\right)$$

After a Laplace transformation, (16) can be expressed as:(17)$$p_{0}\cdot \sigma +p_{1}\cdot \sigma \cdot s+p_{2}\cdot \sigma \cdot s^{2}=\frac{1}{s}\cdot q_{0}\cdot \varepsilon_{0}+q_{1}\cdot \varepsilon_{0}+q_{2}\cdot \varepsilon_{0}\cdot s$$

Solving above equation for σ:(18)$$\sigma =\frac{\varepsilon_{0}\cdot \left(q_{0}\cdot \frac{1}{s}+q_{1}+q_{2}\cdot s\right)}{p_{0}+p_{1}\cdot s+p_{2}\cdot s^{2}}$$

After transformation:(19)$$\sigma =\frac{\varepsilon_{0}\cdot \left(q_{0}\cdot \frac{1}{s}+q_{1}+q_{2}\cdot s\right)}{p_{2}\cdot \left(s-\rho_{1}\right)\cdot \left(s-\rho_{2}\right)}$$
where [19]:(20)$$\rho_{1}=\frac{1}{2\cdot p_{2}}\cdot \left[-p_{1}+\left(p_{1}^{2}-4\cdot p_{2}\cdot p_{0}\right){}^{\frac{1}{2}}\right]$$
(21)$$\rho_{2}=\frac{1}{2\cdot p_{2}}\cdot \left[-p_{1}-\left(p_{1}^{2}-4\cdot p_{2}\cdot p_{0}\right){}^{\frac{1}{2}}\right]$$

Solving (19) using an inverse Laplace transformation, the stress can be expressed as:(22)$$\sigma \left(t\right)=\frac{\varepsilon_{0}}{p_{2}\cdot \rho_{1}\cdot \rho_{2}}\cdot \left\{q_{0}-\frac{1}{\rho_{2}-\rho_{1}}\cdot [\overset{}{\rho_{2}}\cdot e^{\rho_{1}\cdot t}\right.\cdot \left(q_{0}+q_{1}\cdot \rho_{1}+q_{2}\cdot \rho_{1}^{2}\right)-\;\left.\rho_{1}\cdot e^{\rho_{2}\cdot t}\cdot \left(q_{0}+q_{1}\cdot \rho_{2}+q_{2}\cdot \rho_{2}^{2}\right)]\right\}$$

The final form of the model (22) can be solved with known boundary conditions. The solution of the above equation required the use of nonlinear computational methods. For the PUR 80 ShA material, the results of the solutions are summarized in Table 2, assuming a minimization of the mean-square deviations from the measured data as the optimization criterion.

## 4. Discussion

As can be seen from the analysis of the proposed model for 80 ShA, only the parameter *E*_0_ had minor fluctuations from 8.47–13.32 MPa. The other parameters did not show greater deviations from the established values of <1%. It is worth noting that above 50%, fluctuations in the parameter *E*_0_ did not occur. Despite these difficulties, the results obtained can be considered as acceptable, and the model represented it correctly as well. It should be noted that in the adopted assumptions, *E*_0_ was determined for a linear Hooke’s law, even in a very narrow range, and nevertheless, the hyperelastic material may require further modifications based on *E* = *f*(*ε*). The fitting plots of the models to the experimental data are presented below (Figure 8, Figure 9, Figure 10 and Figure 11).

Figures with experimental data fitting to the suggested model (Figure 12, Figure 13, Figure 14 and Figure 15) for the 90 ShA sample are shown below.

Moreover, excellent congruence between the experimental results and the model was observed for 90 ShA, just as it was for 80 ShA before. In addition, the same tendency for fluctuations in elastic constants of model 11.9–15.99 is worth noting, which should be considered as a functional parameter.

## 5. Conclusions

The observable threshold that differentiated the values of the elasticity constants in the model was a 50% strain; it is postulated that this threshold should be kept as a critical quantity in the description of relaxation. Above this value, small fluctuations in the elasticity constants were no longer observed. A rheological description (stress relaxation) of PUR material for different hardness values was proposed based on the formulated five-parameter constitutive model, which was valid in a wide range of strain levels of 25–100% and with particular attention being paid to the large strains above >50%, which complements the existing deficiencies in this description in terms of the constancy of the elastic parameters of the model [8].

## Figures and Tables

**Figure 1 materials-16-03156-f001:**
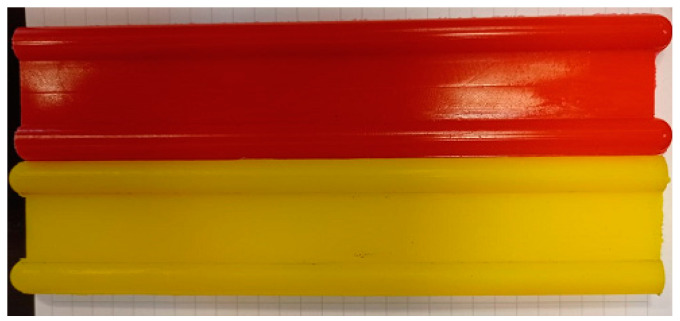
PS (pure shear) specimen geometry (width W = 140 mm, thickness t = 2 mm, effective height h = 15 mm) for relaxation test.

**Figure 2 materials-16-03156-f002:**
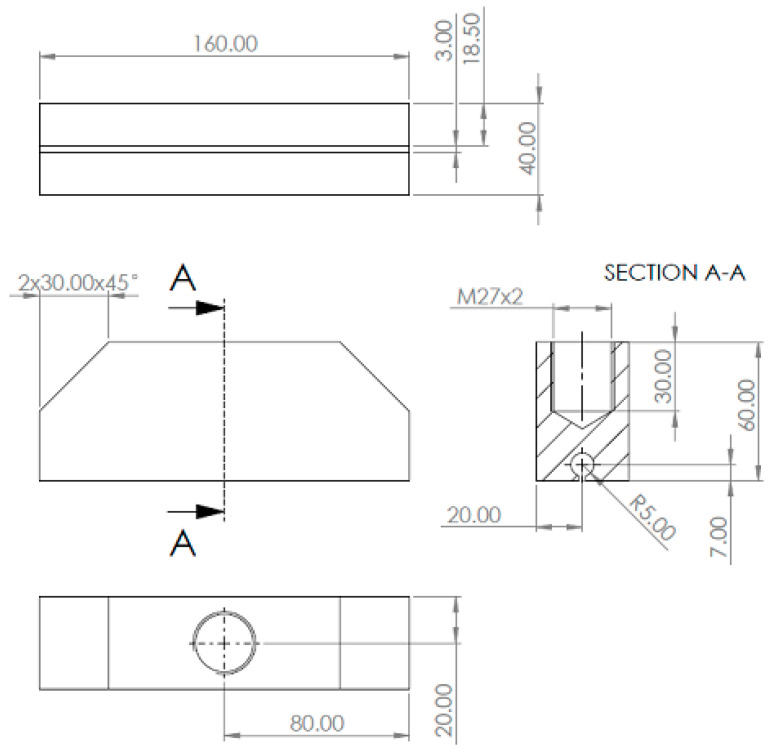
Designed clamping system for tensile uniaxial machine. All dimensions are in mm.

**Figure 3 materials-16-03156-f003:**
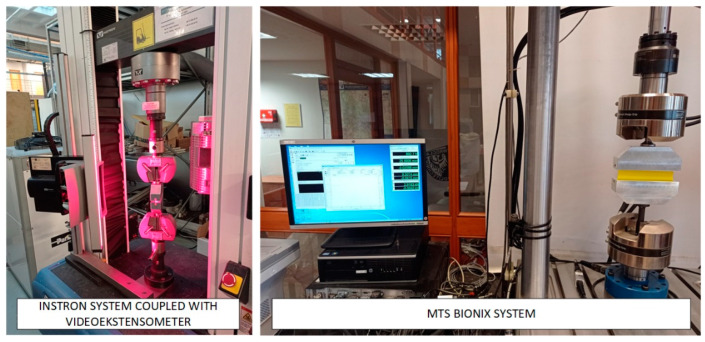
Experimental set-up for pure shear (PS) testing using Instron and MTS test system.

**Figure 4 materials-16-03156-f004:**
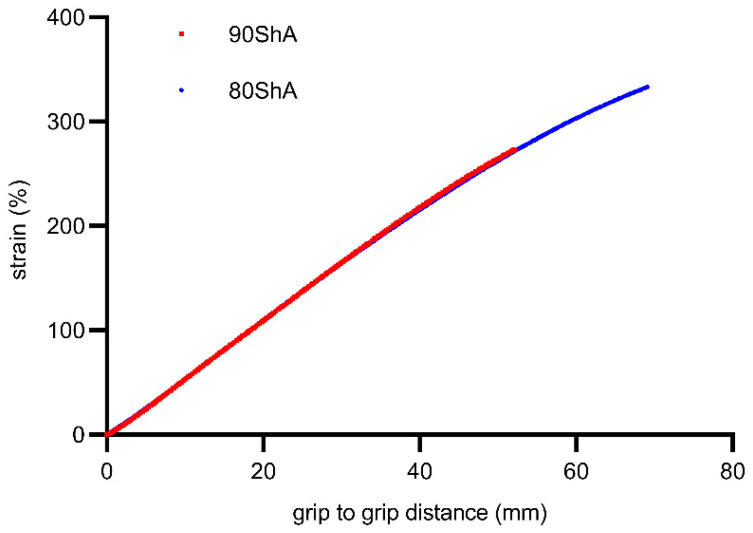
Calibration curves for PS (pure shear) specimens.

**Figure 5 materials-16-03156-f005:**
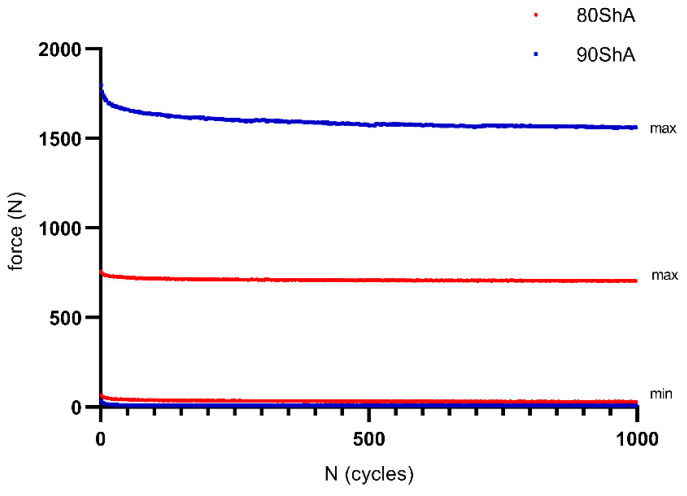
Cyclic response of PS (pure shear) specimens during pre-cycling.

**Figure 6 materials-16-03156-f006:**
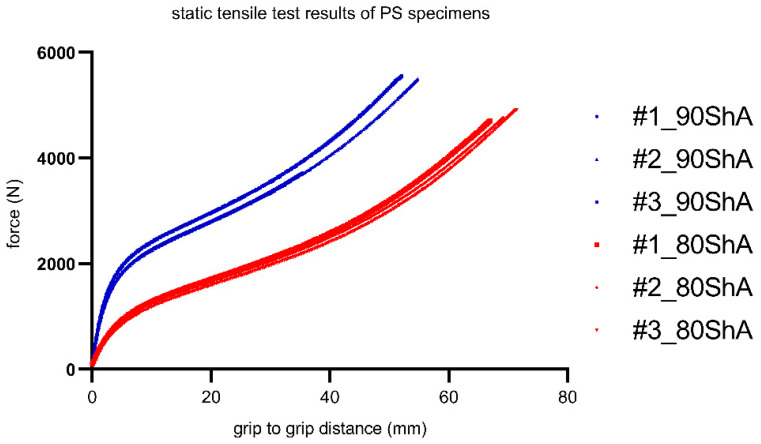
Static load–displacement curves for PS (pure shear) specimens.

**Figure 7 materials-16-03156-f007:**
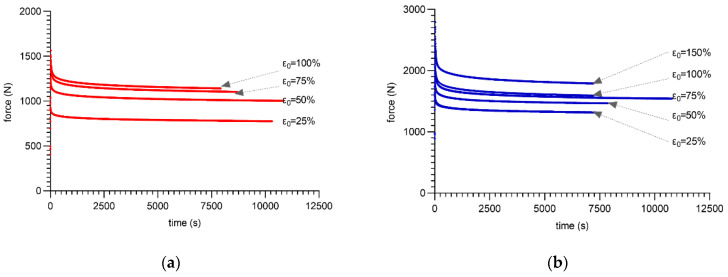
Stress relaxation test—force response: (**a**) mean curves for PUR80; (**b**) mean curves for PUR90.

**Figure 8 materials-16-03156-f008:**
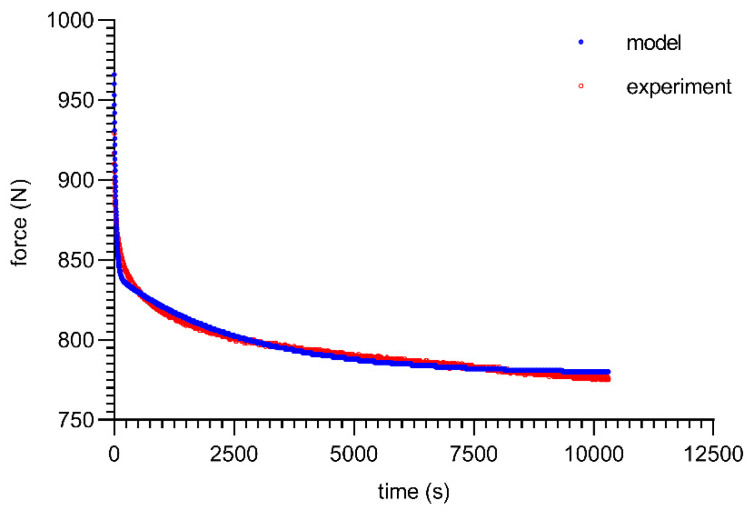
Material 80 ShA—five-parameter relaxation model and experimental data for *ε* = 25%.

**Figure 9 materials-16-03156-f009:**
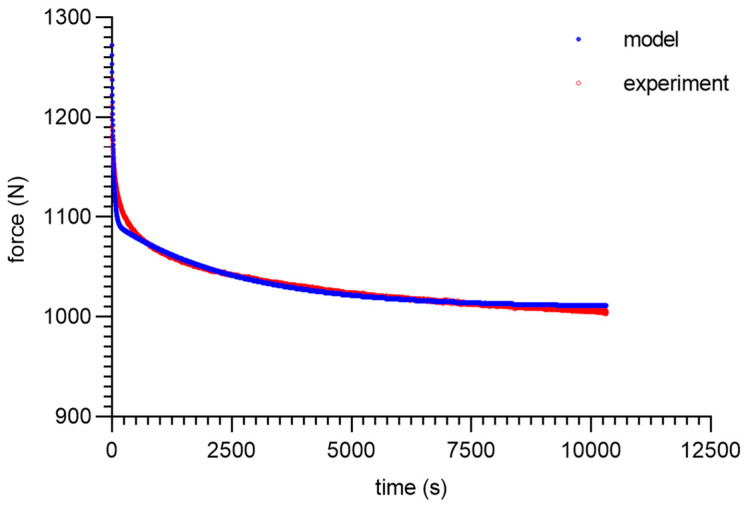
Material 80 ShA—five-parameter relaxation model and experimental data for *ε* = 50%.

**Figure 10 materials-16-03156-f010:**
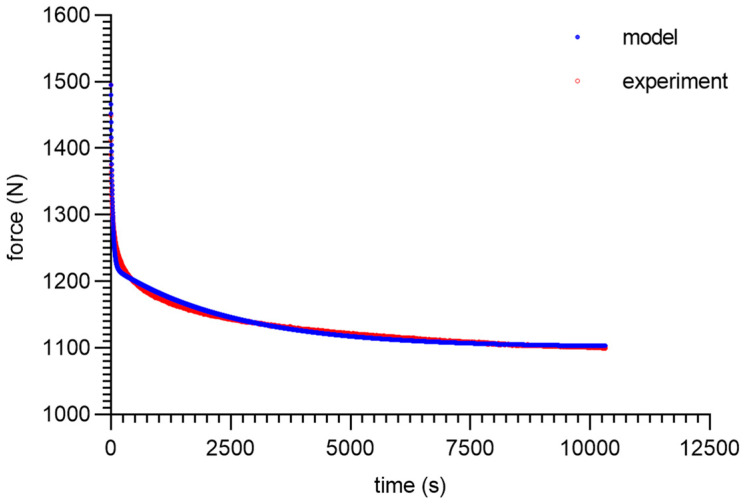
Material 80 ShA—five-parameter relaxation model and experimental data for *ε* = 75%.

**Figure 11 materials-16-03156-f011:**
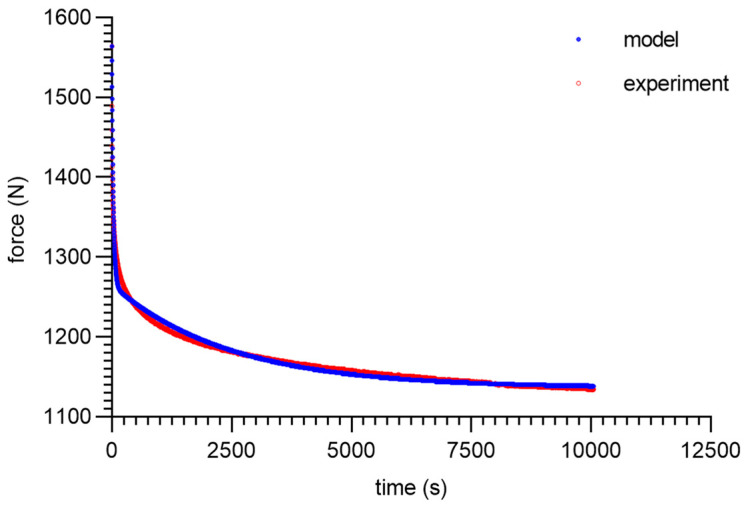
Material 80 ShA—five-parameter relaxation model and experimental data for *ε* = 100%.

**Figure 12 materials-16-03156-f012:**
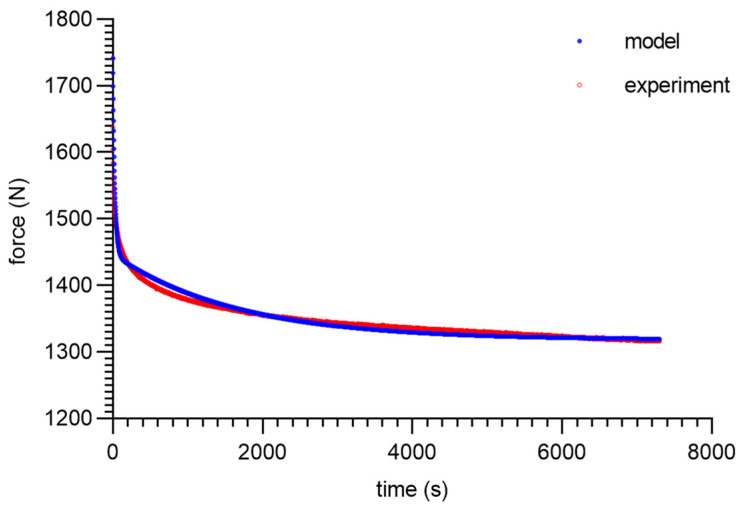
Material 90 ShA—five-parameter relaxation model and experimental data for *ε* = 25%.

**Figure 13 materials-16-03156-f013:**
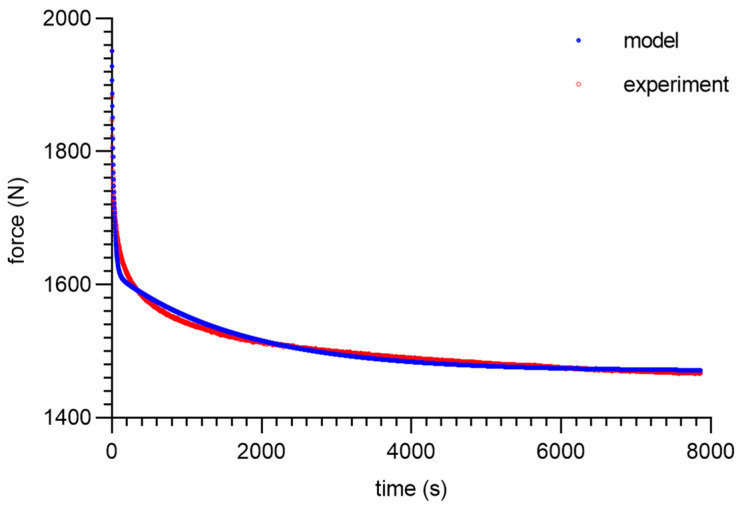
Material 90 ShA—five-parameter relaxation model and experimental data for *ε* = 50%.

**Figure 14 materials-16-03156-f014:**
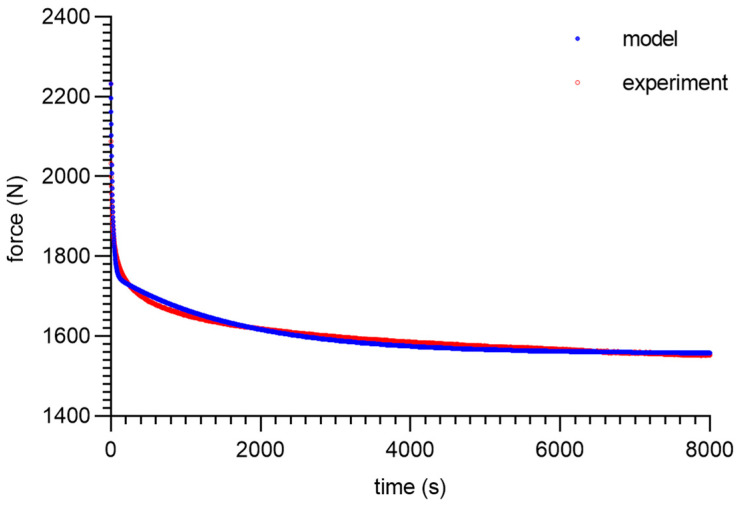
Material 90 ShA—five-parameter relaxation model and experimental data for *ε* = 75%.

**Figure 15 materials-16-03156-f015:**
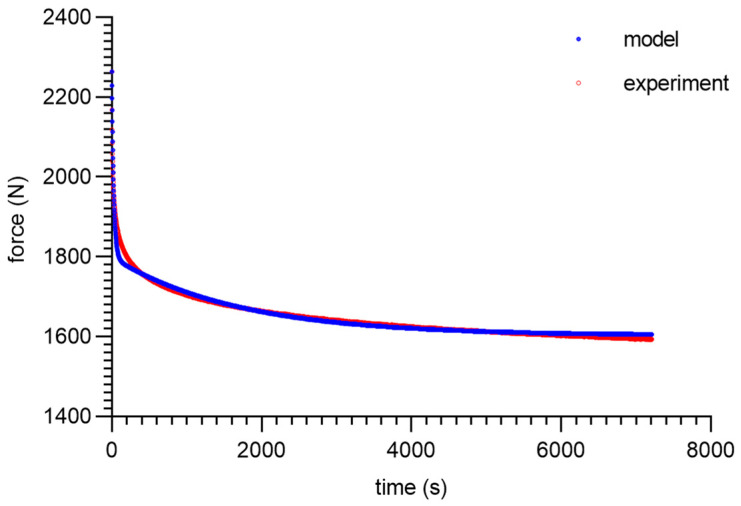
Material 90 ShA—five-parameter relaxation model and experimental data for *ε* = 100%.

**Table 1 materials-16-03156-t001:** Tensile test results analysis for 80 ShA and 90 ShA material configuration based on [21].

Specimen ID	UTS—Ultimate Tensile Strength in MPa	A—Elongation at Break in %
80 ShA	19.4 ± 2.3	710.4 ± 43.9
90 ShA	27.9 ± 0.2	535.3 ± 21.9

**Table 2 materials-16-03156-t002:** Stress relaxation model material data for 80 ShA and 90 ShA.

	*E*_0_ (MPa)	*E*_1_ (MPa)	*E*_2_ (MPa)	*η*_1_ (MPa∙s)	*η*_2_ (MPa∙s)	R^2^
80 ShA-25%	8.47	56.92	92.39	2411.36	259,006.98	0.92
80 ShA-50%	9.17	56.87	92.69	2441.50	259,108.88	0.93
80 ShA-75%	12.64	56.87	92.69	2441.50	259,108.88	0.94
80 ShA-100%	13.32	56.58	94.3	2264.29	258,169.65	0.93
90 ShA-25%	11.91	58.76	100.39	1847.00	181,866.00	0.91
90 ShA-50%	12.09	59.03	98.40	1993.56	183,629.76	0.93
90 ShA-75%	15.99	59.12	97.68	1912.86	184,578.90	0.92
90 ShA-100%	15.21	59.12	97.68	1912.86	184,578.90	0.93

## Data Availability

Data is unavailable.

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
