# Peer review of "Stress Relaxation Behaviour Modeling in Rigid Polyurethane (PU) Elastomeric Materials"

_materials, 2023, doi:10.3390/ma16083156_

Round 1
Reviewer 1 Report
The manuscript entitled “Stress relaxation behaviour modeling in rigid polyurethane (PUR) elastomeric materials” has been reviewed. Detailed comments are as follows:
1. Many typo errors exist in the manuscript. Units should be separated from the numerical value by a space. Please double-check the whole manuscript.
2. In the title and abstract, the abbreviation of polyurethane should be PU, not PUR.
3. The abstract is too short and should be enriched.
4. In the introduction, the necessity and importance of the present work should be stated.
5. In Table 1, (median±std.dev) should be removed.
6. The abbreviations, such as PS, should be named more than once.
7. In the caption of Fig. 3, Intron and should be Instron.
8. In all figures, the first letters of axis names should be capitalized.
9. In Fig. 4, - should be %.
10. In Table 2, the number (1 and 2) for viscosity should be in the form of subscript.
11. For better comparison, Figs. 8-11 should be combined into one figure and it is the same with Figs. 12-15.
12. In References, some journal names, volumes and pages or article numbers are missing. Some article titles are in the capitalization of the first letters.
Author Response
Ms. Ref. No.: materials-2274136
Title: Stress relaxation behaviour modeling in rigid polyurethane (PU) elastomeric materials
Response to Reviewer #1
Dear Reviewer,
I would like to thank for your meticulous review of our manuscript. Thank you for your time and effort.
#R 1. Many typo errors exist in the manuscript. Units should be separated from the numerical value by a space. Please double-check the whole manuscript.
#A Dear Reviewer, thank you for this comment. We have improved it.
#R2. In the title and abstract, the abbreviation of polyurethane should be PU, not PUR.
#A Dear Reviewer. Thank you for this suggestion. We have modified the title and all abbreviations from PUR to PU.
#R3. The abstract is too short and should be enriched..
#A Thank you. This is a very valuable and important recommendation. Please check the latest version of manuscript, here is improved abstract.
#R4. In the introduction, the necessity and importance of the present work should be stated.
#A We fully agree with the reviewer's opinion. After reading the paper, we decided to improve the shape of the introduction.
#R5. In Table 1, (median±std.dev) should be removed.
#A We fully agree with the reviewer's opinion. After reading the paper, we decided to delete confusing phrase.
#R6. The abbreviations, such as PS, should be named more than once.
#A We fully agree with the reviewer's opinion. After reading the paper, we decided to add extension of this abbreviation.
#R7. In the caption of Fig. 3, Intron and should be Instron.
#A We fully agree with the reviewer's opinion. Corrected.
#R8. In all figures, the first letters of axis names should be capitalized.
#A Dear Reviewer. In our graphics system, the accepted standard is how the GRAPHPAD system automatically does the acquisition. Due to limited access to source code and licensing restrictions, we propose to stay with the unchanged form. However, we hope that this style - nota bene acceptable in many journals - will also find acceptance in the eyes of the Reviewer.
#R9. In Fig. 4, - should be %.
#A We fully agree with the reviewer's opinion. Corrected, done.
#R10. In Table 2, the number (1 and 2) for viscosity should be in the form of subscript.
#A Thank you. We also agree with this. However, please note that the font "Symbol", unfortunately looks like this - and indeed 1 and 2 are written in the lower typeface.
#R11. For better comparison, Figs. 8-11 should be combined into one figure and it is the same with Figs. 12-15.
#A Dear Reviewer. It could certainly benefit you. However, the authors intended to accurately illustrate the models' individuating fits to the equation. We are concerned that with the acquisition of all the curves, some of them could disturb the field of observation. Also, keeping in mind the quality of the images, we decided to improve their quality by leaving them unchanged. We firmly believe that the reviewer will appreciate the willingness to show individual indications and will accept.
#R12. In References, some journal names, volumes and pages or article numbers are missing. Some article titles are in the capitalization of the first letters.
#A We fully agree with the reviewer's opinion. After reading the paper, we decided to improve the references. Authors warmly thanks to editorial team, who in case of acceptance for sure will standardize all to MDPI standards.
Looking forward to your favourable consideration
Corresponding author

Reviewer 2 Report
This is an interesting paper, and it is recommended to be accepted for publication after some revision on the basis of comments below.
It is disturbing that two kinds of identifications are used for the samples, that is, PU80 and PU90 in Table 1, and ShA80 and Sha90 in the text and figure captions. One single sample identification coding is suggested.
There should be a space between numbers and units. For instance, correctly t=2 mm (and not 2mm) and h=15 mm (and not 15mm) in the caption of Figure 1, etc. The authors should correct these in the whole manuscript.
Figures 5 and 8-15 should be improved. The axes, the numbers and text along the axes are displayed in bad quality in these figures.
The meaning of the red dots in Figure 4, and the red and blue dots in Figure 6 are unclear. These should either be explained in the text clearly, or these Figures should be modified accordingly, i.e. by deleting these dots.
Figure 4 is not mentioned, not referred in the text of the manuscript.
It is not described in details where the general description, that is, equation (1), comes from. This should be described and discussed properly and in details in the manuscript text.
Author Response
Ms. Ref. No.: materials-2274136
Title: Stress relaxation behaviour modeling in rigid polyurethane (PU) elastomeric materials
Response to Reviewer #2
Dear Reviewer,
I would like to thank for your meticulous review of our manuscript. Thank you for your time and effort.
#R 1. It is disturbing that two kinds of identifications are used for the samples, that is, PU80 and PU90 in Table 1, and ShA80 and Sha90 in the text and Figure captions. One single sample identification coding is suggested.
#A Dear Reviewer, thank you for this comment. We have improved it.
#R2. There should be a space between numbers and units. For instance, correctly t=2 mm (and not 2mm) and h=15 mm (and not 15mm) in the caption of Figure 1, etc. The authors should correct these in the whole manuscript.
#A Dear Reviewer. Thank you. All have been checked and corrected.
#R3. The abstract is too short and should be enriched..
#A Thank you. This is a precious and important recommendation. We have improved it. Please find our last version.
#R4. Figures 5 and 8-15 should be improved. The axes, the numbers and text along the axes are displayed in bad quality in these figures.
#A We fully agree with the Reviewer's opinion. After reading the paper, we decided to improve the quality of all Figures.
#R5. The meaning of the red dots in Figure 4, and the red and blue dots in Figure 6 are unclear. These should either be explained in the text clearly, or these Figures should be modified accordingly, i.e. by deleting these dots.
#A We fully agree with the Reviewer's opinion. Please note that the mentioned Figure is improved. For Reviewer's information – abandoned dots represent individual points registered during the test. After breaking, it was still recorded. Of course, the diagram without points after breaking looks much better – so, again, the Authors' are grateful to Reviewer for such a remark.
#R6. Figure 4 is not mentioned, not referred in the text of the manuscript.
#A Thank you for the information about this mistake. We have improved it.
#R7. It is not described in details where the general description, that is, equation (1), comes from. This should be described and discussed properly and in details in the manuscript text.
#A We fully agree with the Reviewer's opinion. After reading the paper, we decided to improve it, and add some information. However, this model results from modelling using the standard Kelvin-Voigt model [19], and of course, we tried to show a path for the final form. On the other hand, it's also an issue for other theoretical papers – this is more linked with experiments and comparison of models' efficiency for 80ShA and 90 ShA.
Look forward to your favourable consideration
Corresponding author
Reviewer 3 Report
Dear Authors,
I studied your manuscript entitled "Stress relaxation behaviour modeling in rigid polyurethane (PUR) elastomeric materials". Some spaces need to be improved in terms of journal quality. I recommend minor revision before further consideration for publication in the Materials.
1) The abstract and conclusion don't reflect the principal findings of the research title.
2) The introduction is not logically connected and provides limited background on the subject. In addition, this section should be providing a sufficient background of the subject with up-to-date information and the purpose of this study.
3) The text should be reviewed thoroughly to remove some incorrect or confusing phrases.
Author Response
Response to Reviewer #3
Dear Reviewer,
I would like to thank for your meticulous review of our manuscript. Thank you for your time and effort.
#R 1. The abstract and conclusion don't reflect the principal findings of the research title.
#A Dear Reviewer, thank you for this comment. We have improved it.
#R2. The introduction is not logically connected and provides limited background on the subject. In addition, this section should be providing a sufficient background of the subject with up-to-date information and the purpose of this study.
#A Dear Reviewer. After revising the manuscript, we have improved the introduction. All requested changes are marked in yellow colour. We believe that now the paper is more comprehensive organized and enriched.
#R3. The text should be reviewed thoroughly to remove some incorrect or confusing phrases.
#A Thank you. This is a very valuable and important recommendation. Please check the latest version of revised manuscript.
Look forward to your favourable consideration
Corresponding author
Round 2
Reviewer 1 Report
The manuscript has been well revised. It can be accepted if the following comments are considered:
1. In abstract, the full name of PUR should be given.
2. In Line 136, INSTRON should be Instron.
3. In Table 2, the number of viscosities should be in the subscript form.
4. In References, the title of papers, for e.g., Refs. 6 and 10 should not be in the form of the capitalization of first letters.
Author Response
Ms. Ref. No.: materials-2274136-R2
Title: Stress relaxation behaviour modeling in rigid polyurethane (PU) elastomeric materials
Response to Reviewer #1
Dear Reviewer,
I would like to thank for your meticulous review of our manuscript. Thank you for your time and effort.
#R 1. In abstract, the full name of PUR should be given.
#A Dear Reviewer, thank you for this comment. We have improved it.
#R 2. In Line 136, INSTRON should be Instron.
#A Correction has been made.
#R 3. In Table 2, the number of viscosities should be in the subscript form.
#A Correction has been made.
#R4: In References, the title of papers, for e.g., Refs. 6 and 10 should not be in the form of the capitalization of first letters.
#A Correction has been made in mentioned Refs. and remaining issues.
Look forward to your favourable consideration
Corresponding author
